# Analysis of Internal Angle Error of UAV LiDAR Based on Rotating Mirror Scanning

Hao Zhou [1], Qingzhou Mao [1,*], Yufei Song [1], Anlei Wu [2] and Xueqing Hu [1]

1   School of Remote Sensing and Information Engineering, Wuhan University, Wuhan 430079, China
2   Wuhan Luojia Yiyun Photoelectric Technology Co., Ltd., Wuhan 430079, China
*   Correspondence: qzhmao@whu.edu.cn

**Abstract:** UAV LiDAR is a powerful tool for rapidly acquiring ground-based 3D spatial information and has been used in various applications. In addition to the ranging mechanism, the scanning method is also an important factor, affecting the performance of UAV LiDAR, and the internal angle error of LiDAR will seriously affect its measurement accuracy. Starting from the rotary scanning model of a single-sided mirror, this paper presents a comparative study of the characteristics of 45° single-sided mirror scanning, polygon prism scanning, polygon tower mirror scanning, and wedge mirror scanning. The error sources of the quadrangular tower mirror scanning are analyzed in detail, including the angle deviation between the direction of emitted laser and the rotation axis (typical $0.13 \pm 0.18°$ and $0.85° \pm 0.26°$), the angle deviation between the mirror's reflection plane and the rotation axis, and the surface angle deviation between multiple surfaces (typical $\pm 0.06°$). As a result, the measurement deviation caused by the internal angle error can be as high as decimeter to meter, which cannot be fully compensated by simply adjusting the installation angle between the UAV and the LiDAR. After the calibration of the internal angle error, the standard deviation of the elevation difference between the point cloud and the control point is only 0.024 m in the flight experiment at 300 m altitude.

**Keywords:** UAV LiDAR; scanning method; angle error; rotating polygon mirror

## 1. Introduction

Airborne LiDAR is an active detection technology that can quickly acquire the three-dimensional spatial information of ground targets from the air. It is widely used in digital terrain model (DTM) generation [1], classification of power lines [2], acquisition of forest and other vegetation parameters and forest vertical structure parameters [3], coastal terrain mapping [4], disaster relief in dangerous areas [5], 3D urban landscape modeling [6], landcover classification [7], ice surface change monitoring [8], and other fields. In 2004, RIEGL first produced a small spot LiDAR system, LMS-Q560 [9], with full waveform digitization capability, and then successively launched several airborne LiDAR products, such as the VQ-1560 series. In addition, Leica's ALS80 laser scanning system, Teledyne Optech's Galaxy series, also provides a convenient way to carry out large-scale, high-precision, high-density airborne LiDAR point cloud data acquisition [10]. With the development of UAV technology and the miniaturization of airborne LiDAR systems, UAV LiDAR systems are gaining greater advantages over terrestrial laser scanning and mobile laser scanning system, as they are free to the restriction of terrain, and thus capable of obtaining a wider coverage of the scanned region. [11].

For UAV LiDAR system, in addition to ranging mechanism, the scanning method is also an important factor in determining the performance of LiDAR and its field of application [12]. At present, different commercial UAV LiDAR use different scanning methods. For example, RIEGL's VUX1-UAV and miniVUX-2UAV use a rotating 45° mirror for scanning, which can achieve 360° field of view (FOV). Both RIEGL's VUX-240 and

Optech's CL-90 use a rotating polygon mirror scanning method to focus scan lines on ground targets and obtain more scan lines. Both DJI Livox series and Leica's ALS80 use the Risley-prism scanning method [13], which is able to avoid invalid scanning areas and ensure that all laser pulses can be used for effective measurement.

When users use commercial UAV LiDAR to acquire point cloud data, most of them perform point cloud registration [14] by simply adjusting the installation angle between UAV and LiDAR. However, this process assumes that there's no error in the measurement data of the LiDAR (generally, the LiDAR has been calibrated before delivery). Therefore, when there is internal angle error in the LiDAR system, the problem of multi-band point cloud registration cannot be solved by simple adjustment of the installation angle. In this case, the LiDAR scanning method must be taken into consideration and the angle error and its influence must be analyzed to finally complete the registration of point cloud by means of parameter compensation.

The article will first introduce the UAV LiDAR—Luojia Yiyun FT1500, which adopts the scanning method of rotating polygon mirror. Then, we analyze the mathematical model of mirror scanning and derive various scanning modes by setting different parameters in the model. After that, according to the scanning mode of the rotating polygon mirror of Luojia Yiyun FT1500, the existing angle error is analyzed. Finally, the model simulation is used to analyze the influence of each angle error source on the point cloud, and through the actual flight experiment, the difference between the point cloud before and after the angle error compensation is compared. The experiment result clearly shows that if there is internal angle error in the LiDAR system, ignorance of this will seriously affect the accuracy of point cloud registration.

## 2. Materials and Methods

### 2.1. System Composition

The Luojia Yiyun FT1500 developed by us is a UAV LiDAR, which features waveform digitization and online waveform processing and ranging. The maximum measurement can reach up to 2 million points per second, and the measurement range can reach up to 1500 m. It adopts the scanning method of rotating quadrangular tower mirror, which can provide an 80° FOV and can reach as high as 300 scan lines per second. It can be flexibly mounted on a variety of UAV platforms, including multi-rotor UAVs and fixed-wing UAVs, and it is also suitable for light and small-manned helicopters (Figure 1). Key specifications of the Luojia Yiyun FT1500 are listed in Table 1.

**Table 1.** Luojia Yiyun FT1500 system specifications.

| Parameter | Luojia Yiyun FT1500 |
|---|---|
| Wavelength | 1550 nm |
| Measuring range | 1–1500 m @ $\rho \geq 80\%$ |
| Accuracy/Precision | 10 mm/5 mm @ 100 m(1σ) |
| Pulse repetition rate | 100–2000 kHz |
| Maximum number of targets per pulse | up to 7 |
| Laser beam divergence | 0.5 mrad |
| Scanning mechanism | rotating polygon mirror |
| Field of view | 80° |
| Scan speed | up to 300 lines/second |
| Angle measurement resolution | 0.0006° |
| Weight | 2.7 kg |

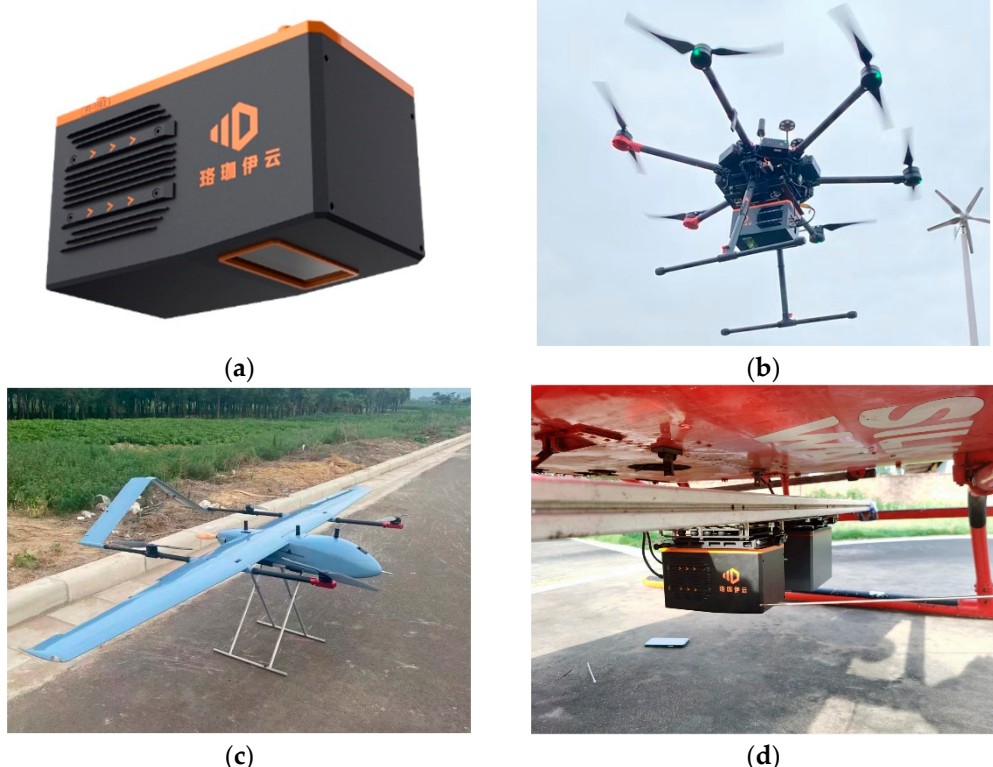

**Figure 1.** Equipment and platform. (**a**) Luojia Yiyun FT1500; (**b**) multi-rotor UAV; (**c**) fixed-wing UAV; (**d**) manned helicopter. The text in the image is the Chinese name (Luojia Yiyun) for LiDAR.

The ranging module and scanning module of the system will be introduced below. The design of the optomechanical structure is shown in Figure 2.

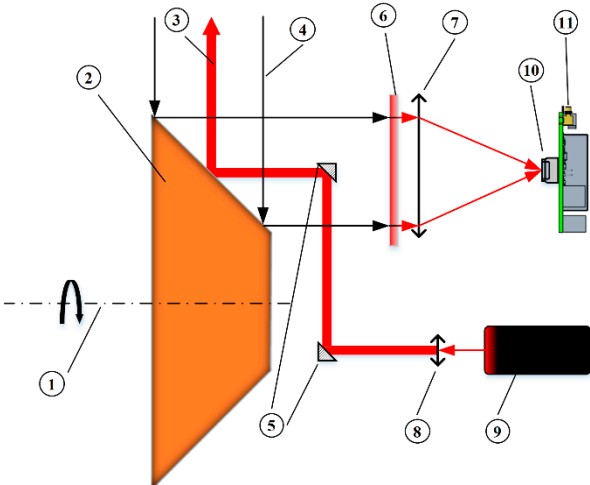

**Figure 2.** The design of the optomechanical structure. 1. Motor rotation axis; 2. Quadrangular tower mirror; 3. Emitted light; 4. Echo light; 5. Reflector; 6. Narrow band filters; 7. Aspheric focusing lens; 8. Collimating lens; 9. Laser; 10. Avalanche photodiode (APD); 11. Analog amplifier circuit.

### 2.1.1. Ranging Module

The ranging module is mainly composed of three parts: laser emitting subsystem, laser receiving subsystem, and online waveform processing subsystem.

1 Laser receiving subsystem

At present, the commonly used laser wavelengths for LiDAR are 905 nm and 1550 nm. In order to achieve a larger measuring range while ensuring human eye safety, we use a

1550 nm fiber laser [15]. The emitted laser beam is split into two proportional parts. The seed light is directly received by the PIN photodiode inside the LiDAR as the starting signal; the main light is reflected by the scanning module after passing through the collimating optical system. The divergence angle of the collimating optical system should be carefully chosen. Small laser divergence angle may make it difficult for the laser beam to detect small objects (such as power lines), while large divergence angle may make laser spots in long distances cover more area, resulting in its energy being too scattered, thus reducing its ability to detect and distinguish between fine objects [16]. After trade-off, a collimating optical system with a divergence angle of 0.5 mrad is adopted.

2    Laser-receiving subsystem

The laser-receiving subsystem is designed to be coaxial with the laser-emitting subsystem to ensure that its FOV corresponds to the laser foot point, thus, avoiding the problem of receiving FOV blindness that exists in non-coaxial systems. The laser echo is filtered by a narrow-band filter to suppress the background light noise at other wavelengths and is then converged into the detector's sensitive area by the receiving optical system. The use of aspheric mirror can ensure that the echo energy is focused on the detector's photo-sensitive area while reducing the number of components used and the space occupied by the receiving optical system. InGaAs APD is used to detect light at 1550 nm wavelength, as APD can obtain higher gain and signal-to-noise ratio, compared with PIN photodiode, which makes it more suitable for long-distance weak signal detection. The APD converts the received light signal into a current signal, which is subsequently converted into a voltage signal through trans-impedance amplifier (TIA). The output weak voltage signal of the TIA is further amplified by a low-noise amplifier to match the ADC's operating input voltage range.

Echo power $P_r$ reflected by the Lambertian surface and received by the detector can be expressed as follows [17]:

$$P_r = \frac{P_t \eta_t \eta_r \eta_{at} A_r \rho}{4\pi R^2} \tag{1}$$

where $P_t$ is the emitted laser power, $\eta_t$ is the optical efficiency of the laser emitting optical system, $A_r$ is the effective receiving area of the optical system, $\rho$ is the target reflectivity, $\eta_{at}$ is the attenuation coefficient of the laser in the atmosphere, and $R$ is the target distance. The echo power is inversely proportional to the square of the target distance. The larger the dynamic range of the ranging, the larger the dynamic range of the echo power. The general echo amplification circuit has a fixed gain and linear amplification interval, so it cannot realize the linear amplification of both long-distance and short-distance echoes at the same time. It will inevitably lead to either saturation distortion of the short-distance echo waveform or cause a low signal-to-noise ratio for the long-distance echo waveform. Moreover, the ADC also has its own voltage input range. Only when the amplitude of the amplified waveform is within the ADC voltage input range, can the analog signal be converted into a digital signal without being truncated. For the reasons mentioned above, multi-channel amplification is used to obtain electrical signals with different magnifications of the same echo signal to ensure that the signals at far and near distances can be within a reasonable range [4]. In order to obtain higher ranging precision, higher sampling frequency and narrower pulse width are required [18]. A 4-channel ADC with 1.25 GHz sampling rate is used to digitize the seed signal and echo signal of nanosecond pulse width at high speed.

3    Online waveform processing

Our LiDAR adopts the direct time-of-flight (dToF) method as the ranging mechanism. The ranging is achieved by measuring the detection time difference of the emitted laser and the echoed laser. By compressing the energy into shorter pulses, the dToF method can obtain larger instantaneous power, so it has great advantages in long-distance detection.

The system will record the detection time of the emitted laser $t_0$ and detection time of echoed laser $t_r$, and the ranging value $R$ of the measured target can be expressed as:

$$R = \frac{c(t_r - t_0)}{2} \tag{2}$$

where c is the speed of light. The waveform obtained by high-speed ADC can be used for full-waveform analysis, in which the digitized echo signal is first stored and then processed by a post-processing algorithm to extract underlying waveform information from sampled data. As many waveform processing algorithms contain time-consuming operations, they cannot be implemented in real time in LiDAR systems [19]. However, many LiDAR operators and users usually do not need this underlying waveform information. They only use the ranging and reflectivity information of the targets so their core requirement has a higher ranging accuracy and higher measurement frequency. For this reason, what is indispensable for the new LiDAR capable of waveform digitization is an online waveform processing module that can generate ranging values in real time and at high frequency [20]. The high-speed ADC transmits the digitized waveform to the high-speed waveform processing unit, which extracts effective pulse data from massive digitized values and obtains pulse amplitude and flight time in real time. By implementing efficient algorithms in hardware, the Luojia Yiyun FT1500 can achieve a maximum measurement frequency of 2000 kHz and process up to 7 consecutive echoes for a single emitted pulse.

### 2.1.2. Scanning Module

For UAV LiDARs, some commonly used scanning methods include 45° single-sided rotating mirror scanning, polygon prism scanning, polygon tower mirror scanning, wedge mirror circular scanning, and so on.

In terms of the distribution of generated point cloud, 45° single-sided rotating mirror scanning and polygon tower mirror scanning can obtain scanning point clouds along straight and evenly distributed scan lines. The scanning track obtained by polygon prism scanning and wedge mirror scanning are non-linear and unevenly distributed, resulting in the heterogeneity of point densities in scanned region.

In terms of effective scanning region, 45° single-sided mirror scanning can obtain a 360° FOV. However, in UAV-related applications, most desired targets are in the region below the UAV, so a FOV of 360° will cause more than half of the scan lines in invalid region. Polygon prism scanning, polygon tower mirror scanning, and wedge mirror scanning can limit the effective scanning angle to only cover the ground target, reducing invalid scanning region and producing more scan lines at the same motor speed.

In order to obtain a point cloud with more scan lines and evenly distributed scanning track, polygon tower mirror scanning is an ideal scanning method for UAV LiDAR.

### 2.2. Mirror Scanning Model

These scanning methods mentioned above, such as 45° single-sided mirror scanning, polygon prism scanning, polygon tower mirror scanning, and wedge mirror scanning, actually share the same underlying mathematical model. Their differences just lie in the way the scanning region is divided by the mirror's different reflection planes, the angle between the laser emission direction and the rotation axis, and the angle between the reflection plane and the rotation axis. For example, a polygon tower mirror can be regarded as a composition of multiple single-sided mirrors, so, in the following, the scanning model of a single-sided mirror will be first analyzed.

### 2.2.1. Single-Sided Mirror Scanning Model

As shown in Figure 3, the X-axis is the motor rotation axis, XOY is the plane with a rotation angle of 0, the Z-axis is determined by the right-hand rule, $S(x_s, y_s, z_s)$ is the laser emitting point, $P_0$ is the plane of the single-sided mirror, $\vec{d}$ is the unit normal vector of the plane, $M(m, 0, 0)$ is the intersection of the reflection plane and the X-axis, $\varphi$ is the

angle between the reflection plane and the Y-axis, $R$ is the reflection point where the laser intersects the reflecting surface, $A$ is the coordinate of the target in the point cloud, and $\theta$ is the rotation angle. The laser emitting point $S$ and the intersection point $M$ can be determined by structural design parameters, since they only affect the translation and not the angle.

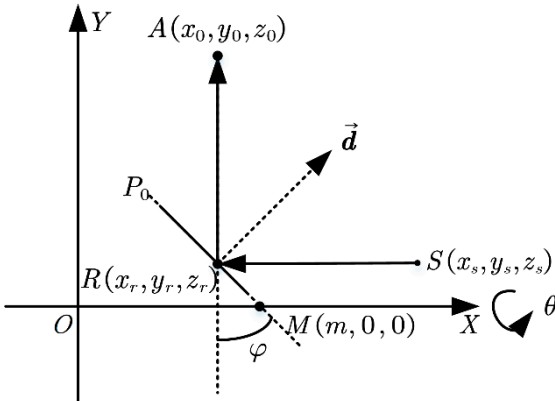

**Figure 3.** Schematic diagram of coordinate calculation of single-sided mirror scanning model.

The unit normal vector $\overrightarrow{d}$ of the reflection plane can be expressed as:

$$\overrightarrow{d} = (\cos \varphi, \cos \theta \sin \varphi, \sin \theta \sin \varphi) \tag{3}$$

The reflection plane $P_0$ can be expressed as:

$$\overrightarrow{d} \cdot (x, y, z) = m \cos \varphi \tag{4}$$

The reflection point $R$ can be expressed as:

$$\begin{bmatrix} x_r \\ y_r \\ z_r \end{bmatrix} = \begin{bmatrix} x_s \\ y_s \\ z_s \end{bmatrix} + \left| \overrightarrow{SR} \right| R_z R_y \begin{bmatrix} -1 \\ 0 \\ 0 \end{bmatrix} \tag{5}$$

$$R_y = \begin{bmatrix} \cos \omega_y & 0 & \sin \omega_y \\ 0 & 1 & 0 \\ -\sin \omega_y & 0 & \cos \omega_y \end{bmatrix} \tag{6}$$

$$R_z = \begin{bmatrix} \cos \omega_z & -\sin \omega_z & 0 \\ \sin \omega_z & \cos \omega_z & 0 \\ 0 & 0 & 1 \end{bmatrix} \tag{7}$$

In the above equation, the unit vector along laser emission direction is obtained by rotating the unit vector along the negative direction of X-axis around Y-axis and Z-axis in turn; $\omega_y$ and $\omega_z$ are the rotation angle and $R_y$ and $R_z$ are the rotation matrices, respectively. $\left| \overrightarrow{SR} \right|$ can be obtained from Equations (4) and (5).

The reflected ray unit vector is:

$$\begin{bmatrix} r_x \\ r_y \\ r_z \end{bmatrix} = R_r R_z R_y \begin{bmatrix} -1 \\ 0 \\ 0 \end{bmatrix} \tag{8}$$

where $\boldsymbol{R}_r$ is the reflection matrix of the reflection plane $P_0$ [21]:

$$\boldsymbol{R}_r = \begin{bmatrix} -\cos 2\varphi & -\sin 2\varphi \cos \theta & -\sin 2\varphi \sin \theta \\ -\sin 2\varphi \cos \theta & 1 - 2\sin^2 \varphi \cos^2 \theta & -\sin^2 \varphi \sin 2\theta \\ -\sin 2\varphi \sin \theta & -\sin^2 \varphi \sin 2\theta & 1 - 2\sin^2 \varphi \sin^2 \theta \end{bmatrix} \tag{9}$$

The coordinate of target A in the space rectangular coordinate system is:

$$\begin{bmatrix} x_0 \\ y_0 \\ z_0 \end{bmatrix} = \left( r - \left| \overrightarrow{SR} \right| \right) \begin{bmatrix} r_x \\ r_y \\ r_z \end{bmatrix} + \begin{bmatrix} x_r \\ y_r \\ z_r \end{bmatrix} \tag{10}$$

where $r$ is the LiDAR ranging value.

Obviously, according to the above model analysis, the characteristics of 45° single-sided mirror scanning are: single-sided mirror, laser emission direction parallel to the rotation axis, a 45° angle between the reflection plane, and the rotation axis. This is also one of the most used scanning methods.

### 2.2.2. Polygon Prism and Polygon Tower Mirror Scanning Model

Both the polygon prism and the polygon tower mirror are equivalent to a combination of multiple single-sided mirrors, and the intention is to divide the 360° of the motor rotation to maximize the use of all angles. In order not to cause ambiguity, in this paper, polygon prism refers to the kind of rotating mirror whose reflection planes are parallel to its central axis, and polygon tower mirror refers to the kind of rotating mirror whose four reflection planes have the same angle between them and the axis. Different scanning tracks and scanning modes can be obtained by changing the angle between the laser emitting direction and the rotation axis.

4　　Quadrangular prism scanning mode 1

As shown in Figure 4a, the reflection plane is parallel to the rotation axis, i.e., $\varphi = 90°$, and the laser emission direction is perpendicular to the rotation axis, i.e., $\omega_z = 90°$. It is a typical quadrangular prism scanning mode. Its scanning schematic diagram and scanning track are shown in Figure 4. Let $\varphi = 90°$ and $\omega_z = 90°$ in Equation (8); the unit vector along the direction of the reflected laser in this scanning method can be obtained as $(0, \cos 2\theta, \sin 2\theta)$. Its X-coordinate is 0, so the scanning track is a straight line, and the scanning angle is theoretically twice the rotation angle $\theta$ of the motor. However, in fact, in the transceiver coaxial system, in order to avoid the situation in which the laser is reflected back into the LiDAR system, the laser emission will be forbidden in certain angle intervals. As a result, rotation angles in those intervals do not correspond to any emitted pulse, so the scanning angle cannot reach the theoretical 180°. In addition, since the scan angle is twice the rotation angle $\theta$, as $\theta$ increases, the separation between points in the same scan line will also increase, resulting in the sparse distribution of point clouds at both ends of the scan line.

To solve the problem of the emitted laser being reflected back into the LiDAR in coaxial system, the laser emitting direction can be designed to be not parallel with the scanning plane, so that the reflected light does not have collinearity with the emitted light. To achieve this, there are two adjustment schemes: adjusting the laser emission angle or adjusting the angle between the reflection surface and the rotation axis.

5　　Quadrangular prism scanning mode 2

The reflection plane is parallel to the rotation axis, i.e., $\varphi = 90°$, and the angle between the laser emission direction and the rotation axis is adjusted to $0° < \omega_z < 90°$. Generally, the angle between the rotation axis and the ground surface is fixed to ensure that the scanning direction is directly below the LiDAR. Taking $\omega_z = 45°$ as an example, the scanning schematic diagram and scanning track are shown in Figure 5. This scanning method solves the problem of collinearity between the emitted light and the reflected light.

As $\omega_z$ gets smaller, the curvature of the scanning track gets larger and the field of view gets narrower; therefore, the sparseness of the edge point cloud is improved.

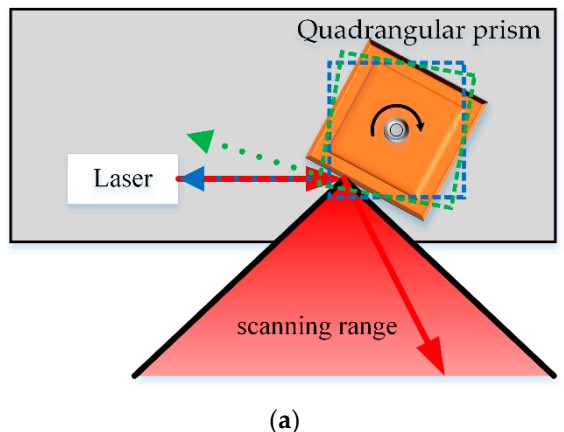

(**a**)

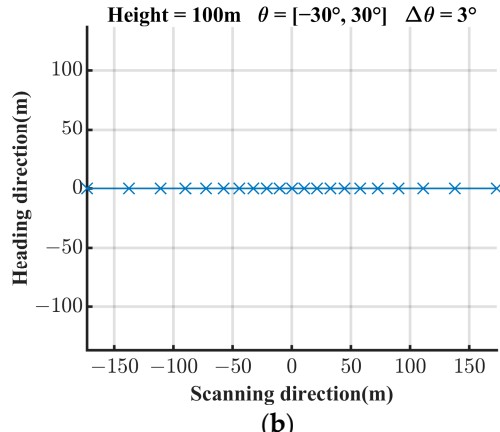

(**b**)

**Figure 4.** Quadrangular prism scanning mode 1. (**a**) Scanning schematic diagram; (**b**) Scanning track with a height of 100 m, a motor rotation angle of $\pm 30°$ and an interval angle of $3°$ between points.

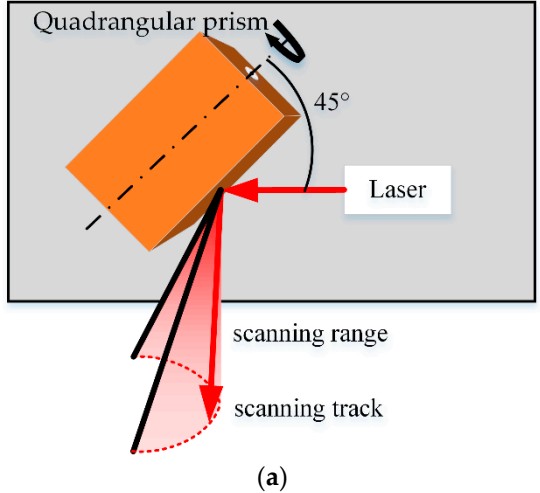

(**a**)

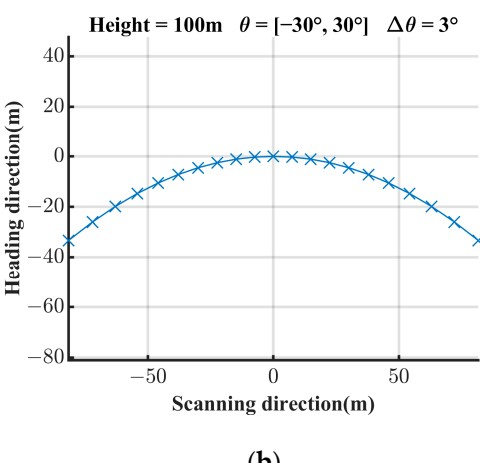

(**b**)

**Figure 5.** Quadrangular prism scanning mode 2. (**a**) Scanning schematic diagram; (**b**) Scanning track with a height of 100 m, a motor rotation angle of $\pm 30°$ and an interval angle of $3°$ between points.

6    Quadrangular tower mirror scanning mode 1

In this mode, the angle between the reflection plane and the rotation axis is adjusted to $0° < \varphi < 90°$, and the laser emission direction is perpendicular to the rotation axis, i.e., $\omega_z = 90°$; it is a scanning method of the tower mirror. When $\varphi = 45°$, the scanning schematic diagram and scanning track are shown in Figure 6. Its FOV is smaller than that of quadrangular prism scanning mode 2, and the curvature of the scanning trajectory is larger.

7    Quadrangular tower mirror scanning mode 2

Another tower mirror scanning method is similar to a $45°$ single-sided mirror, the difference is that it uses four $45°$ mirrors. The angle between the reflection plane and the rotation axis is $\varphi = 45°$, and the laser emission direction is parallel to the rotation axis, i.e., $\omega_z = 0°$. Its scanning schematic diagram and scanning track are shown in Figure 7. The scanning track is consistent with the scanning track of $45°$ single-sided mirror. Using this scanning method, one scan line can be generated for every $90°$ rotation angle of the motor. Therefore, this scanning method has great advantages in UAV LiDAR, and Luojia Yiyun FT1500 uses this scanning method.

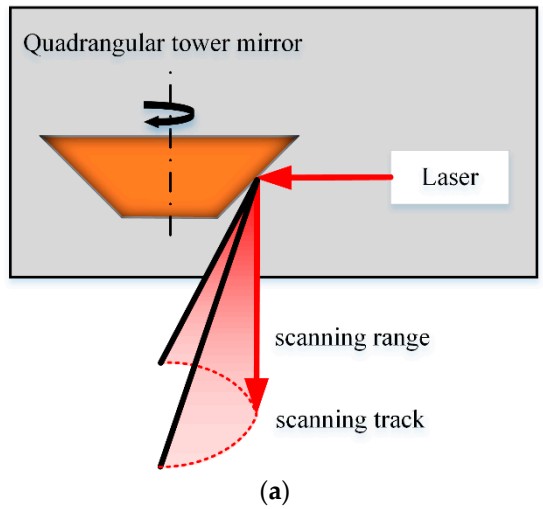

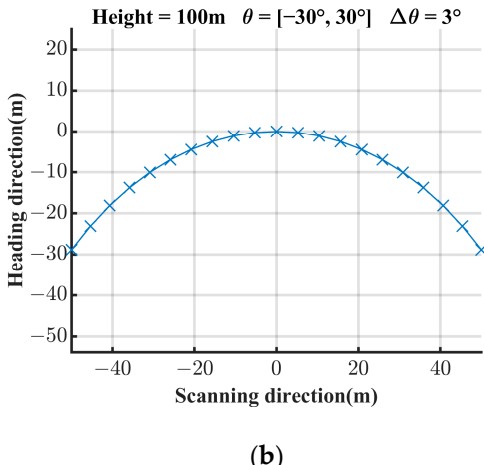

**(a)**

**(b)**

**Figure 6.** Quadrangular tower mirror scanning mode 1. (**a**) Scanning schematic diagram; (**b**) Scanning track with a height of 100 m, a motor rotation angle of $\pm 30°$ and an interval angle of $3°$ between points.

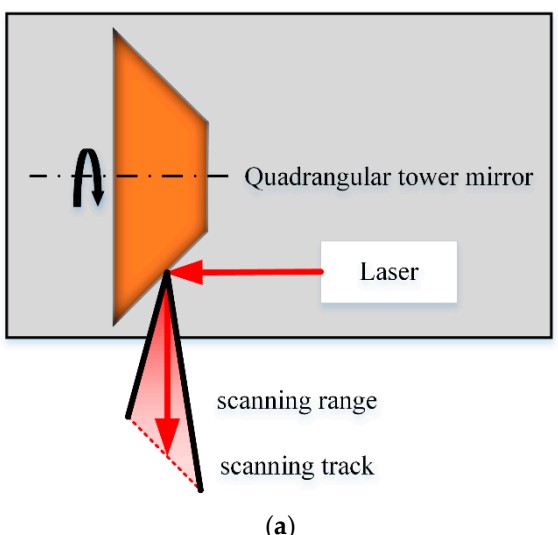

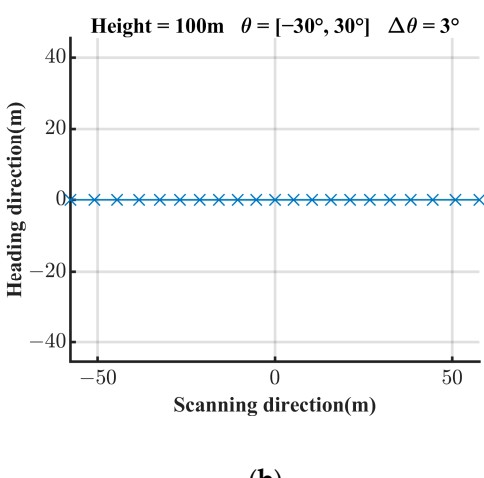

**(a)**

**(b)**

**Figure 7.** Quadrangular tower mirror scanning mode 2. (**a**) Scanning schematic diagram; (**b**) Scanning track with a height of 100 m, a motor rotation angle of $\pm 30°$ and an interval angle of $3°$ between points.

For polygon prism and polygon tower mirror, the laser spot will be split at the junction of the two surfaces, which will cause the scanning angle to be invalid. In addition, the larger the scanning angle is, the smaller the effective receiving area will be, resulting in weaker echo energy, which will affect the signal-to-noise ratio (SNR) and measurement range. Therefore, in practical applications, the scanning angle of the polygon prism and the polygon tower mirror cannot reach the theoretical level.

### 2.2.3. Wedge Mirror Scanning

The wedge mirror scanning also belongs to the single-sided mirror scanning model. By designing a small angle $\varphi$, the scanning surface changes from a flat surface to a conical surface, thereby obtaining a elliptic scanning track. On this basis, a reasonable $\omega_z$ is designed to ensure that the laser is outside the scanning window. Taking $\varphi = 5°$, $\omega_z = 45°$ as an example, the scanning schematic diagram and scanning track are shown in Figure 8. The wedge mirror scanning can truly ensure that all the rotation angles of the motor are

effective and the scanning direction is toward the ground target, which is impossible for the polygon prism and the polygon tower mirror scanning. Since the scanning track is elliptic, the point cloud will be unevenly distributed, and when $\omega_z$ is not 0, the elliptic track will also be deformed.

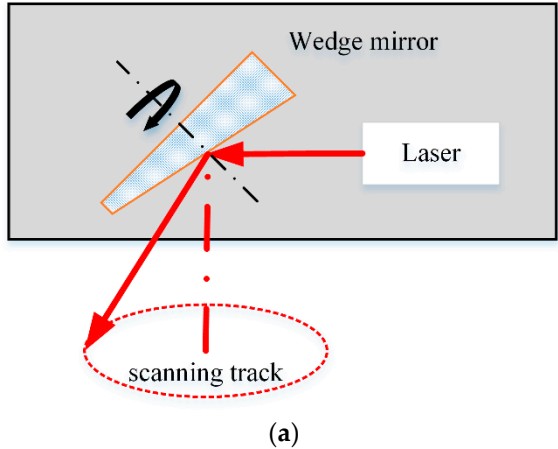

(**a**)

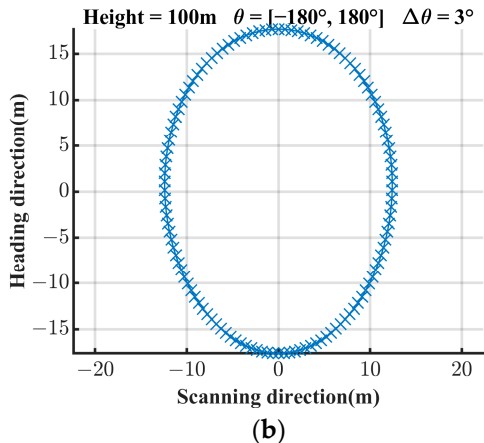

(**b**)

**Figure 8.** Wedge mirror scanning mode. (**a**) Scanning schematic diagram; (**b**) Scanning track with a height of 100 m, a motor rotation angle of $\pm180°$ and an interval angle of $3°$ between points.

### 2.3. Angle Errors of Mirror Scanning Model

Through the single-sided mirror scanning model, we can find that the coordinates of the target point in the space rectangular coordinate system are determined by several angles: angles between laser emission direction and rotation axis, i.e., $\omega_y$ and $\omega_z$; motor rotation angle, i.e., $\theta$; angles between the mirror and the rotation axis, i.e., $\varphi$. The manufacturing and assembly process will inevitably cause certain deviations in these angles. When such deviations are large, there will be a non-negligible influence between the real coordinates of the target and the coordinates obtained by calculation.

#### 2.3.1. Laser and Rotation Axis Parallelism Deviation

During the production process of the equipment, ideally, the laser beam is completely parallel to the rotation axis, but, in fact, the laser beam may be reflected by multiple mirrors. Due to the limitation of manufacturing precision and installation accuracy, there is a non-zero angle between the laser emission direction and rotation axis, which makes the actual emission direction of the laser deviate from the ideal emission direction.

#### 2.3.2. Eccentricity Error of Circular Grating Rotary Encoder

The rotation angle of the tower mirror is obtained by a high-precision photoelectric rotary encoder. When the angle measurement center of the photoelectric encoder does not coincide with the rotation center, an eccentricity error will occur, as shown in Figure 9. The angle measurement deviation between the real rotation angle $\theta$ and the measured rotation angle $\theta'$ by the encoder can be approximately expressed as:

$$\Delta\theta = \theta - \theta' \approx E\sin(\theta' - \theta_e) + E\sin\theta_e \tag{11}$$

$$E = \frac{e}{R} \tag{12}$$

where $E$ is the eccentricity rate between the angle measuring center of the photoelectric encoder and the rotation axis, $e$ is the eccentric distance, $R$ is the distance between the reading head of the photoelectric encoder and the rotation center, and $\theta_e$ is the angle between the eccentric direction and the direction of the reading head.

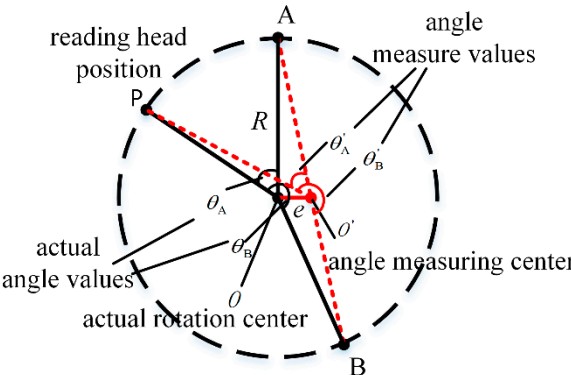

**Figure 9.** Schematic diagram of eccentric error.

The eccentricity error can be compensated by calibration, or it can be directly compensated by installing two reading heads in opposite directions along the encoder's diameter. The opposite installation makes the angles between the two reading heads and the eccentric direction are $\theta_e$ and $\theta_e + 180°$, respectively, that is, $\Delta\theta$ is opposite to each other, so:

$$\theta = \frac{\theta'_A + \Delta\theta_A + \theta'_B + \Delta\theta_B}{2} = \frac{\theta'_A + \theta'_B}{2} \tag{13}$$

Using more mathematical derivation, it can be proved that the eccentric error after the compensation of the opposite double reading heads can be reduced to the order of $E^2$, which is negligible for this system.

### 2.3.3. Surface Angle Deviation

For the quadrangular tower mirror, the machining accuracy is difficult to ensure that the angle between each reflection surface and the bottom is the same as the design value. Therefore, the actual scanning track of different reflection surfaces are no longer the same plane but four planes with included angles. In addition, the projection angles of the normal vectors of different planes on the rotation plane will also have a certain deviation, and they are no longer perpendicular or parallel to each other, as shown in Figure 10. These two errors will simultaneously cause the ground features on the ground scanned by four different planes to be misaligned in the point cloud. Therefore, for each surface of the tower mirror, the surface angle deviations $\Delta\theta_1$, $\Delta\theta_2$, $\Delta\theta_3$, $\Delta\theta_4$ and $\Delta\varphi_1$, $\Delta\varphi_2$, $\Delta\varphi_3$, $\Delta\varphi_4$ need to be calibrated separately. Furthermore, since the Y-axis is not perpendicular to the emission window, a certain angular offset $\theta_0$ is required to make the Y-axis perpendicular to the emission window, and the offset $\theta_0$ can be combined with $\Delta\theta_1$.

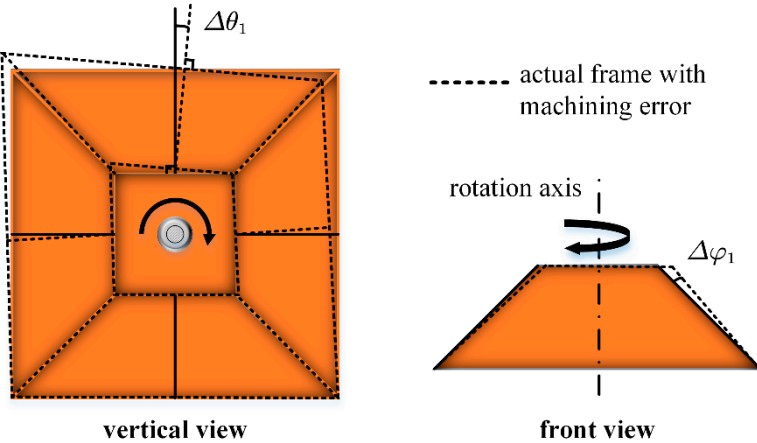

**Figure 10.** Surface angle deviation of quadrangular tower mirror.

## 3. Results

The angle error of tower mirror scanning will seriously affect the overlap and accuracy of the whole point cloud, and these angle errors cannot be compensated by conventional strip adjustment in the form of installation angle. Therefore, it is necessary to reduce the angle error through accurate adjustment or correct each point in the point cloud with calibrated parameters to ensure accuracy. In order to intuitively show the influence of the angle error of the tower mirror scanning on the point cloud, some quantitative simulation experiments and actual flight experiment will be conducted below.

### 3.1. Simulation Experiments

3.1.1. The Effect of $\omega_z$ on the Point Cloud

As shown in Figure 11, flight 1 is in the same direction as flight 2, and flight 3 is coincidental with flight 2 but in the opposite direction. There is a feature point $A$ directly below flight 1, and a feature point $B$ directly below flight 2. In the presence of a non-zero $\omega_z$ in flight 1, the LiDAR scans two points, $A$ and $B$, and obtains their distance and angle, respectively. The positions of the two points $A$ and $B$ in the point cloud calculated after ignoring $\omega_z$ are $A_1'$ and $B_1'$. Similarly, the positions of points $A$ and $B$ in the point cloud are calculated as $A_2'$, $B_2'$, $A_3'$, and $B_3'$ in flight 2 and flight 3.

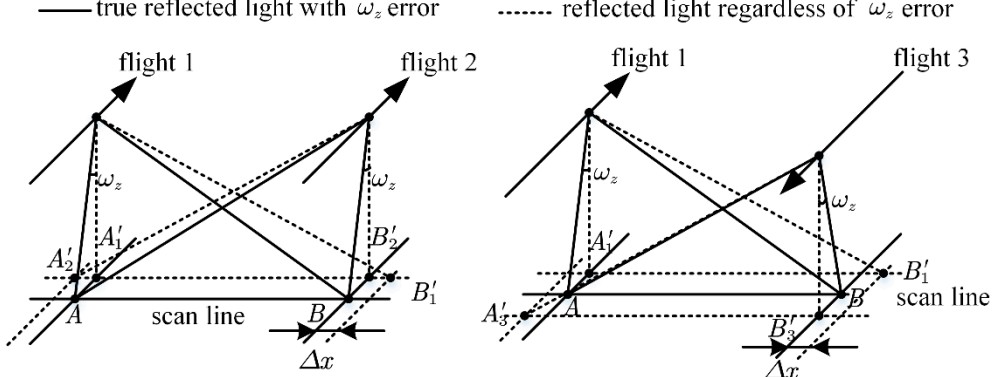

**Figure 11.** Schematic diagram of the simulated flights and scanned objects.

We found that $A_2'$ and $A_3'$ have deviations in the heading direction. By adjusting the installation angle of the pitch direction, it seems possible to ensure that $A_1'$, $A_2'$, $A_3'$ overlap in the heading direction. The key problem is that there is a vertical heading offset $\Delta x$ between $A_1'$ and $A_2'$, and it is the same situation between $B_1'$ and $B_2'$. Due to this offset $\Delta x$, $A_1'$, $A_2'$ and $B_1'$, $B_2'$ in flight 1 and flight 2 cannot be overlapped by adjusting the installation angle of the roll direction,.

The following part will quantitatively analyze the variation of $\Delta x$, $\Delta y$, and $\Delta z$ with the error angle $\omega_z$ and the distance $L$ between the object and the flight directly below. The established simulation model is shown in Figure 12: as flight 1 is set, the y-axis of the ground coordinate system is parallel to the flight direction of the flight 1, the z-axis is the elevation direction, the x-axis is determined by the right-hand coordinate system, flight 1 is on the yoz-plane, point $P$ is the ground point, $L$ is the distance from $P$ to yoz-plane, and point $F$ is the origin of the LiDAR coordinate system, and its position and attitude are determined by six external elements. If there is an error angle, $\omega_z$, when the LiDAR scans the ground point $P$, the coordinates of the LiDAR on the flight are $F(0, t, H)$, and the ranging and angle obtained by the LiDAR are $r_P$ and $\theta_P$. Since the error angle $\omega_z$ is ignored, the coordinates of point P in the point cloud calculated by $r_P$ and $\theta_P$ are $P'$, which will generate deviations $\Delta x$, $\Delta y$, and $\Delta z$ in the directions of the three coordinate axes.

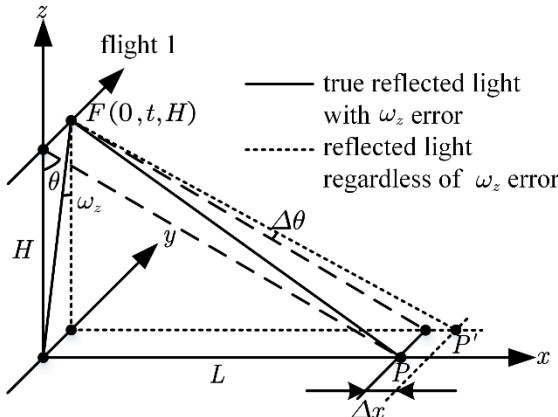

**Figure 12.** Schematic diagram of the simulation model of $\omega_z$.

For the offsets caused by angle errors, the flight height $H$ is a positive proportion factor that converts the angle errors into the translation errors; so in order to display the magnitude of the offsets, we set $H$=100 m. The simulation process and steps are as follows:

8      Change $\omega_z$ or the distance $L$ from point $P$ to the yoz-plane;

9      Use the Gauss–Newton method to solve the coordinate $F(0, t, H)$ on the flight when the LiDAR scans to the point $P$, as well as the ranging $r_P$ and the angle $\theta_P$ obtained by the LiDAR at this time;

10     According to the influence of $\omega_z$ on the point cloud, use $\alpha_p = -\omega_z$ to rotate the point cloud in the pitch direction, and then make $\omega_z = 0$, so as to ensure that the ground objects in flight 2 and flight 3 in Figure 11 coincide. Thus, the process of correcting the point cloud of multi-strip by using the installation angle when $\omega_z$ is ignored is simulated. Then, by $F(0, t, H)$, $r_P$ and $\theta_P$, the coordinates of point $P$ in the point cloud are calculated as $P'$;

11     Finally, $\Delta x$, $\Delta y$, $\Delta z$, etc., are calculated from the coordinates of $P$ and $P'$.

The simulation results are shown in Figure 13:

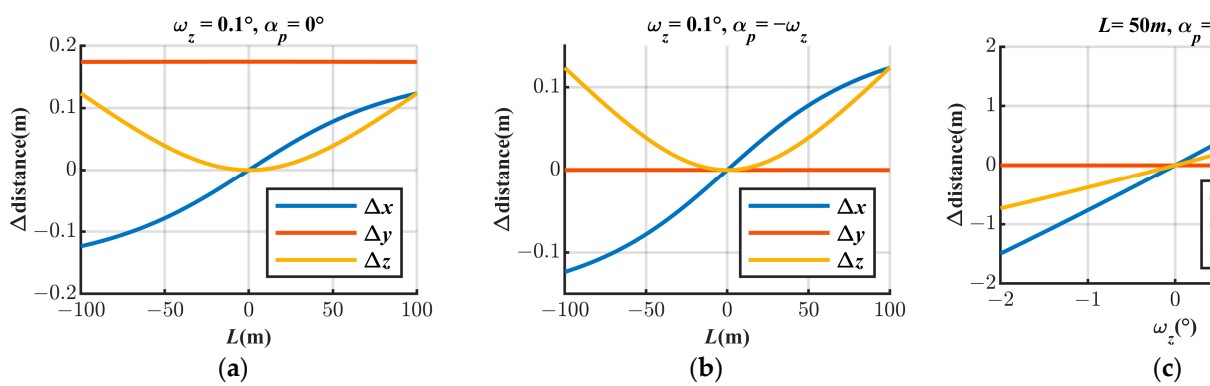

**Figure 13.** The effect of $\omega_z$ on the point coordinates. (**a**) Without $\alpha_p$ compensation, $\Delta x$, $\Delta y$, $\Delta z$ change with $L$; (**b**) After using pitch angle $\alpha_p$ compensation, when $\omega_z = 0.1°$, $\Delta x$, $\Delta y$, $\Delta z$ change with $L$; (**c**) After using pitch angle $\alpha_p$ compensation, when $L = 50$ m, $\Delta x$, $\Delta y$, $\Delta z$ change with $\omega_z$.

The effects of $L$ and $\omega_z$ on the $\Delta y$ of $P$ and $P'$ in the heading direction are almost negligible after the pitch angle correction. There is a large deviation value in the two directions of $\Delta x$ and $\Delta z$. When $H = 100$ m and $\omega_z = 0.1°$, the $\Delta x$ of the object 50 m away from the flight has been deviated by about 8 cm, and the $\Delta z$ has been deviated by about 4 cm. As $L$ and $\omega_z$ increase, $\Delta x$ and $\Delta z$ will increase accordingly, where the direction of $\Delta x$ is related to both the positive and negative of $L$ and $\omega_z$, while the direction of $\Delta z$ is only related to the positive and negative of $\omega_z$. As shown in Figure 13b, under the changing trend of $\Delta x$ and $\Delta z$, it means that the object deviation is symmetrical in the left and right

directions of the flight. That is to say, the point clouds of flight 2 and flight 3 in Figure 11 can overlap, but the point cloud of flight 1 cannot overlap with the point cloud of flight 2 or flight 3.

### 3.1.2. The Effect of $\omega_y$ on the Point Cloud

Similarly, referring to the above analysis of the effect of $\omega_z$ on the point cloud, we continue to use the same simulation model. Different from the effect of $\omega_z$, the effect of $\omega_y$ on the point cloud is approximated by applying the same angle of rotation in the roll direction and the heading direction, as shown in Figure 14.

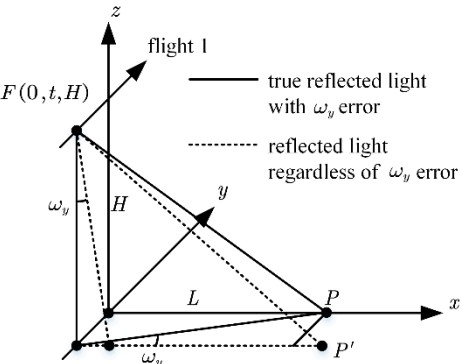

**Figure 14.** Schematic diagram of the simulation model of $\omega_y$.

Therefore, the roll angle $\alpha_r$ and the heading angle $\alpha_h$ will be used for compensation to simulate the situation of using the installation angle to compensate the point cloud of multiple flights when $\omega_y$ is ignored. The simulation results are shown in Figure 15:

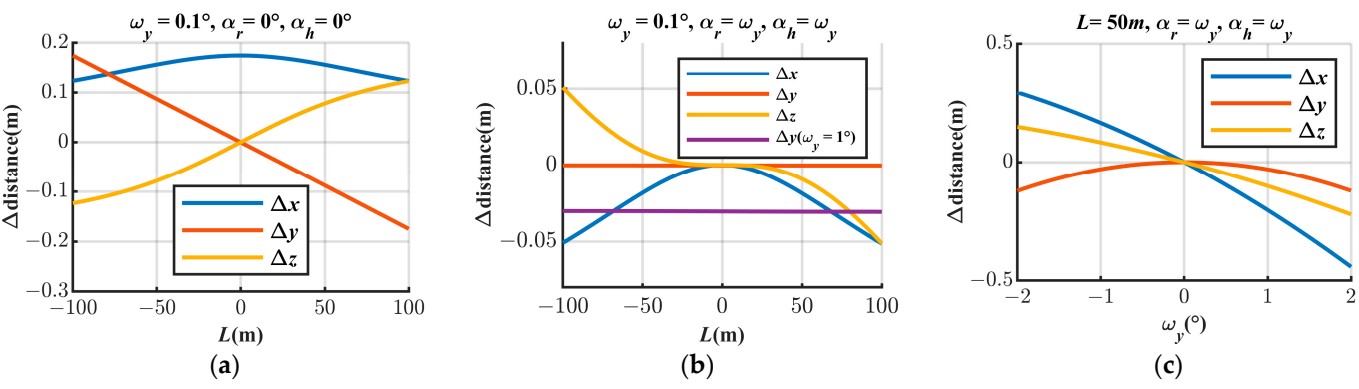

**Figure 15.** The effect of $\omega_y$ on the point coordinates. (**a**) Without $\alpha_r$ and $\alpha_h$ compensation, $\Delta x$, $\Delta y$, $\Delta z$ change with $L$; (**b**) After using roll angle $\alpha_r$ and heading angle $\alpha_h$ compensation, when $\omega_y = 0.1°$, $\Delta x$, $\Delta y$, $\Delta z$ change with $L$. In order to show the change trend of $\Delta y$, a change curve of $\Delta y$ when $\omega_y = 1°$ is added; (**c**) After using roll angle $\alpha_r$ and heading angle $\alpha_h$ compensation, when $L = 50$ m, $\Delta x$, $\Delta y$, $\Delta z$ change with $\omega_y$.

As shown in Figure 15, after the roll and heading angle compensation, the effect of $L$ on the $\Delta y$ of $P$ and $P'$ in the heading direction is negligible. $\Delta y$ mainly increases with the increase in $\omega_y$, and the direction of $\Delta y$ deviation is a fixed negative direction. When $H = 100$ m and $\omega_y = 0.1°$, the $\Delta x$ of the object 50 m away from the flight has been deviated by about 2 cm and the $\Delta z$ has been deviated by about 1 cm. As $L$ and $\omega_y$ increase, $\Delta x$ and $\Delta z$ will increase accordingly, where the direction of $\Delta x$ is only related to the positive and negative of $\omega_y$, while the direction of $\Delta z$ is related to both the positive and negative of $L$ and $\omega_y$. As shown in Figure 15b, under the change trend of $\Delta x$ and $\Delta z$, it means that the deviation of the object is asymmetric in the left and right directions of the flight, and the point clouds cannot completely coincide with each other in the same or opposite flight. If

the roll angle is used forcibly to adjust the opposite flight, and a certain feature is adjusted to coincide, the features in other areas will be layered again.

### 3.1.3. The Effect of $\Delta\varphi$ on the Point Cloud

Intuitively, $\Delta\varphi$ will cause an included angle between the LiDAR scanning plane and the plane of angle $\theta$, so the scanning track will no longer be a straight line, but a curve with a certain radian, as shown in Figure 16. Similarly, the point cloud will also be rotated by a pitch angle, however, the rotation is about twice as large as $\Delta\varphi$ because $\Delta\varphi$ is the angular error of the reflecting surface.

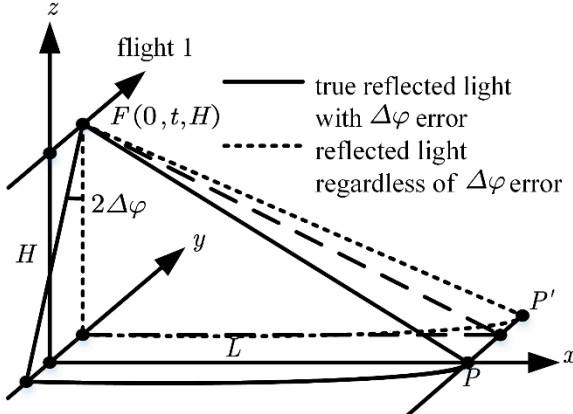

**Figure 16.** Schematic diagram of the simulation model of $\Delta\varphi$.

The pitch angle $\alpha_p$ will be used for compensation to simulate the situation of using the installation angle to compensate the point cloud of multiple flights when $\Delta\varphi$ is ignored. The simulation results are shown in Figure 17:

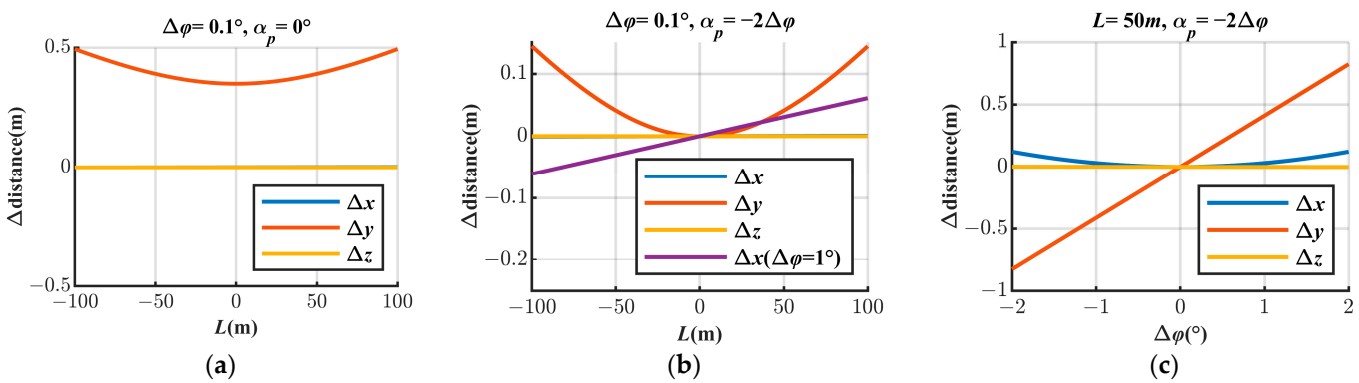

**Figure 17.** The effect of $\Delta\varphi$ on the point coordinates. (**a**) Without $\alpha_p$ compensation, $\Delta x$, $\Delta y$, $\Delta z$ change with $L$; (**b**) After using pitch angle $\alpha_p$ compensation, when $\Delta\varphi = 0.1°$, $\Delta x$, $\Delta y$, $\Delta z$ change with $L$. In order to show the change trend of $\Delta x$, a change curve of $\Delta x$ when $\Delta\varphi = 1°$ is added; (**c**) After using pitch angle $\alpha_p$ compensation, when $L = 50$ m, $\Delta x$, $\Delta y$, $\Delta z$ change with $\Delta\varphi$.

The effects of $L$ and $\Delta\varphi$ on the $\Delta z$ of $P$ and $P'$ in the elevation direction are almost negligible after the pitch angle correction. The deviation in the $\Delta x$ direction is very small, but there is a large deviation in the $\Delta y$ direction. When $H = 100$ m and $\Delta\varphi = 0.1°$, the $\Delta y$ of the object about 50 m away from the flight has already deviated by about 4 cm, $\Delta x$ is negligible. As $L$ and $\Delta\varphi$ increase, $\Delta x$ and $\Delta y$ will increase accordingly, where the direction of $\Delta x$ is only related to the positive and negative of $L$, while the direction of $\Delta y$ is only related to the positive and negative of $\Delta\varphi$. As shown in Figure 17b, under the changing trend of $\Delta y$, it means that the object deviation is symmetrical in the left and right directions of the flight, but the point clouds of the three flight in Figure 11 cannot overlap. Compared

with $\omega_z$, $\Delta\varphi$ has much less influence on $\Delta x$ and $\Delta z$, but mainly affects $\Delta y$, which is also the difference between $\omega_z$ and $\Delta\varphi$, so these two error angles cannot be confused.

For the quadrangular tower mirror, each surface has its own reflection surface angle error, which are $\Delta\varphi_1$, $\Delta\varphi_2$, $\Delta\varphi_3$, and $\Delta\varphi_4$. Since the pitch direction installation angle of the LiDAR is only one, it cannot compensate for $\Delta\varphi_1$, $\Delta\varphi_2$, $\Delta\varphi_3$, and $\Delta\varphi_4$ of four different reflecting surfaces. It can be seen from Figure 17a that when $H = 100$ m and $\Delta\varphi = 0.1°$, the $\Delta y$ of the object directly below the flight has been deviated by about 35 cm. Therefore, it is necessary to calibrate the angle error of the reflecting surface of each surface, otherwise the point clouds of the four reflecting surfaces of the single flight will not be able to overlap.

### 3.2. Flight Experiment

The flight experiment was conducted in Baoxie Town, Wuhan, China, on August 16, 2022, with sunny weather and a temperature of 37 °C. The target area contains feature targets, such as spiked houses, roads, and woods in different directions, which can well demonstrate the stitching effect, measurement accuracy, and other indicators of the multi-strip point cloud. In this flight experiment, a 160 cm axis distance multi-rotor UAV was used, the flight height was set to 300 m, the flight speed was about 8 m/s, and four flight lines were set as shown in Figure 18. The Y-axis in the point cloud pointed to the north direction and the Z-axis pointed to the elevation direction. The scanning speed is 300 lines/s, and the laser trigger frequency is set to 300 kHz.

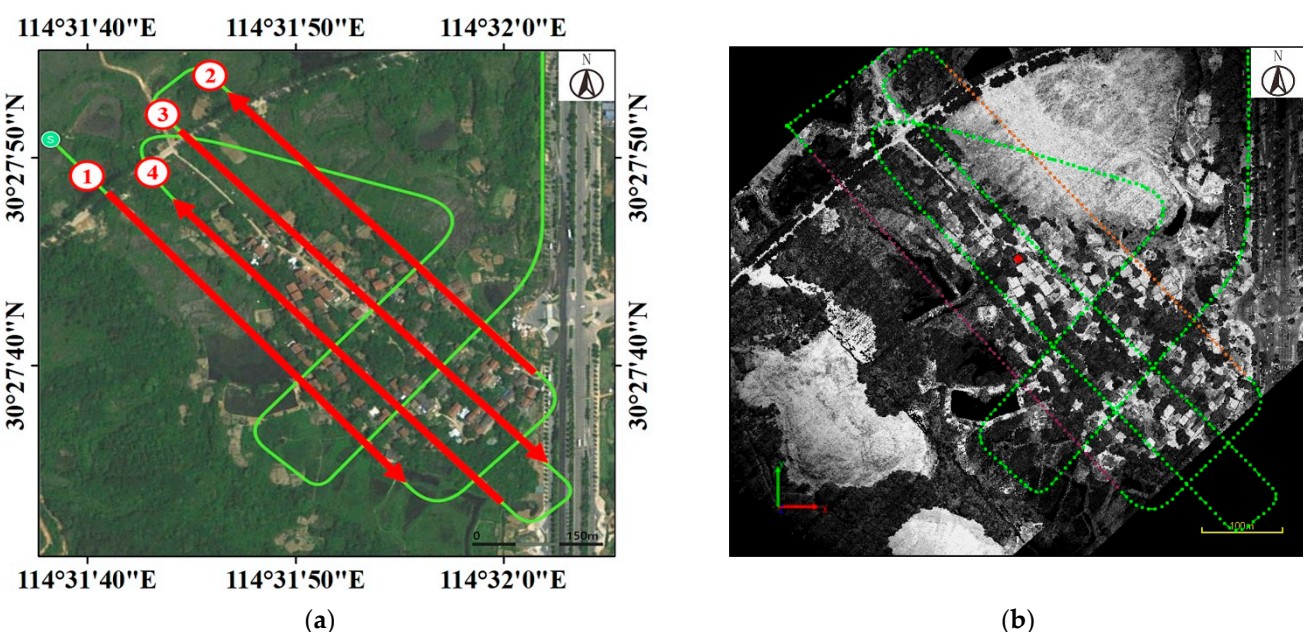

(**a**)  (**b**)

**Figure 18.** Flight experiment area and flight lines diagram. (**a**) UAV planning route and map view; (**b**) Actual flight path and acquired point cloud.

To visualize the effect of angle errors on the point cloud, we have calibrated all angular errors and installation errors between the UAV and the LiDAR in advance, and the resulting point cloud is used as the reference point cloud. The angular error model in this paper is verified by setting a certain angular error parameter to 0 and comparing the difference in point clouds after using installation angle compensation. We have counted the calibrated angle errors of 36 LiDARs of the same type, and the mean and standard deviation of all kinds of angle errors, and the angle errors of the equipment used for flight experiments are shown in Table 2.

**Table 2.** Statistics of calibrated angle errors for LiDARs.

| Parameter | Mean | Std($\sigma$) | The Equipment Used for Flight Experiment |
|:---:|:---:|:---:|:---:|
| $\omega_z$ | $-0.13611°$ | $0.17958°$ | $-0.04000°$ |
| $\omega_y$ | $0.84864°$ | $0.26278°$ | $1.01500°$ |
| $\Delta\theta_2$ | $-0.00653°$ | $0.06561°$ | $-0.02314°$ |
| $\Delta\theta_3$ | $-0.00933°$ | $0.07006°$ | $0.00616°$ |
| $\Delta\theta_4$ | $0.00198°$ | $0.03897°$ | $0.00557°$ |
| $\Delta\varphi_2$ | $0.00293°$ | $0.05526°$ | $0.01656°$ |
| $\Delta\varphi_3$ | $0.02077°$ | $0.06028°$ | $0.01806°$ |
| $\Delta\varphi_4$ | $0.00637°$ | $0.05194°$ | $0.01969°$ |

After making $\omega_z = 0$, as shown in Figure 19, the point cloud localization of two opposing flight strips (flight line 1 and flight line 2) are intercepted parallel to the flight lines (hereafter referred to as parallel section), and it can be found that the deviation (about 40 cm) generated by $\omega_z$ can be compensated by adjusting the pitch angle. As shown in Figure 20, the point cloud localization of two opposing flight strips intercepted perpendicular to the flight lines (hereafter referred to as vertical sections), and Figure 21 shows the vertical sections of the two isotropic flight strips (flight line 2 and flight line 4). It can be found that the point clouds of the vertical sections of both the isotropic and opposite flight directions produce an offset (about 20 cm). However, the offset of the isotropic strips cannot be compensated by adjusting the roll angle.

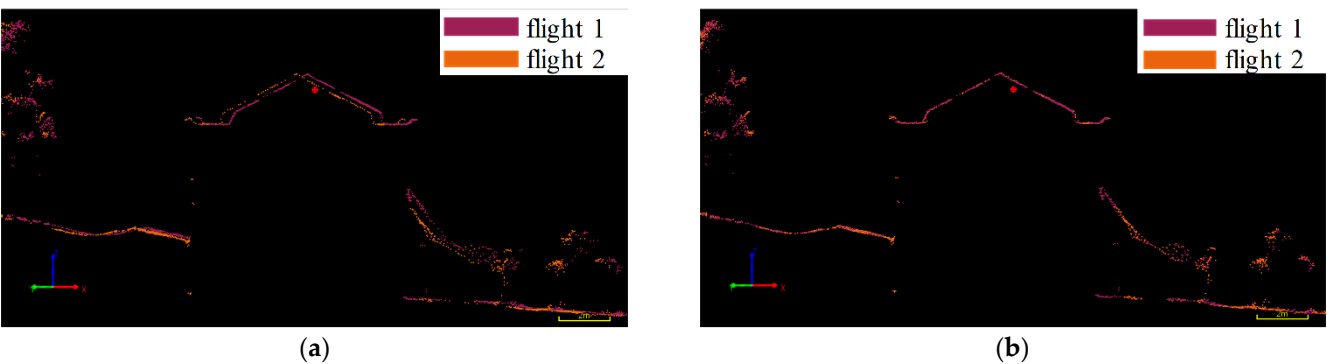

(**a**)                                        (**b**)

**Figure 19.** Parallel section of point cloud in in opposing flight strips (flight 1 and flight 2). (**a**) Without installation angle compensation; (**b**) With installation angle compensation.

After letting $\omega_y = 0$, the vertical section of the two opposing flight strips (flight 2 and flight 3) are intercepted from the point cloud, as shown in Figures 22 and 23. It can be found that the targets with different scanning angles in Figure 23 have different deviations (maximum up to 1 m), which obviously cannot be compensated by adjusting the installation angle alone either.

As shown in Figure 24, when there is a surface angle deviation, the point cloud of the single flight will be layered (about 10 cm to 20 cm), which naturally cannot be compensated by adjusting the installation angle.

Finally, Table 3 shows the elevation difference between the point cloud and real time kinematic (RTK) control points after compensating for the angular error in this flight experiment at 300 m flight height (excluding the point pt13, which is obviously deviated more), and the standard deviation of the elevation difference is 0.024 m.

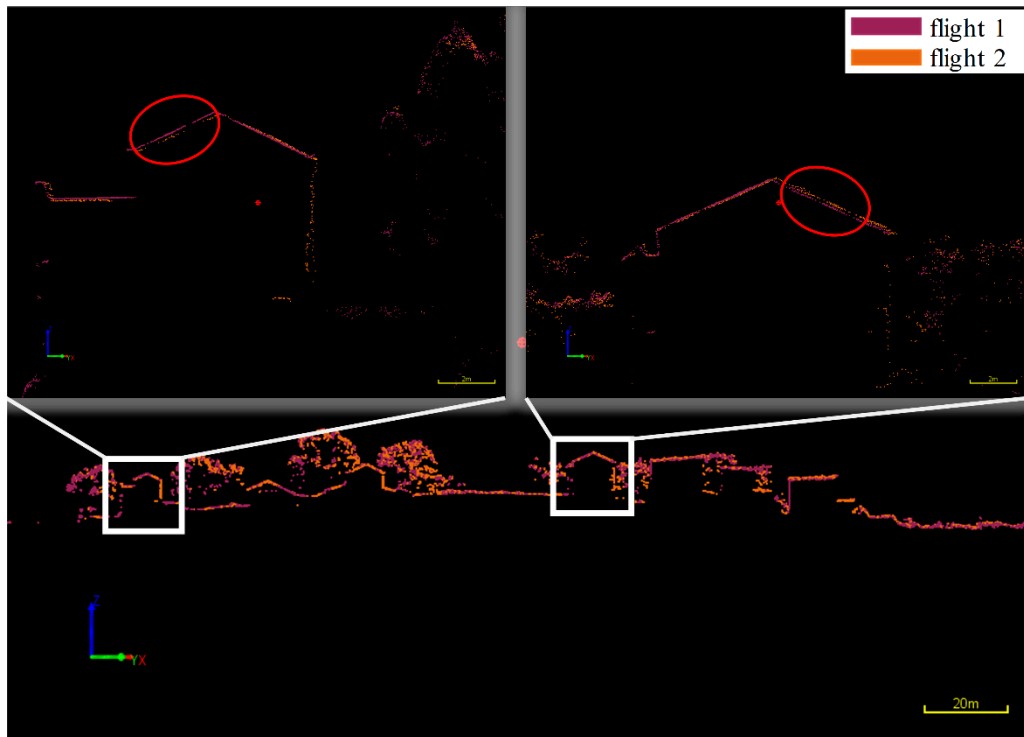

**Figure 20.** Vertical section of point cloud in opposing flight strips (flight 1 and flight 2) with installation angle compensation.

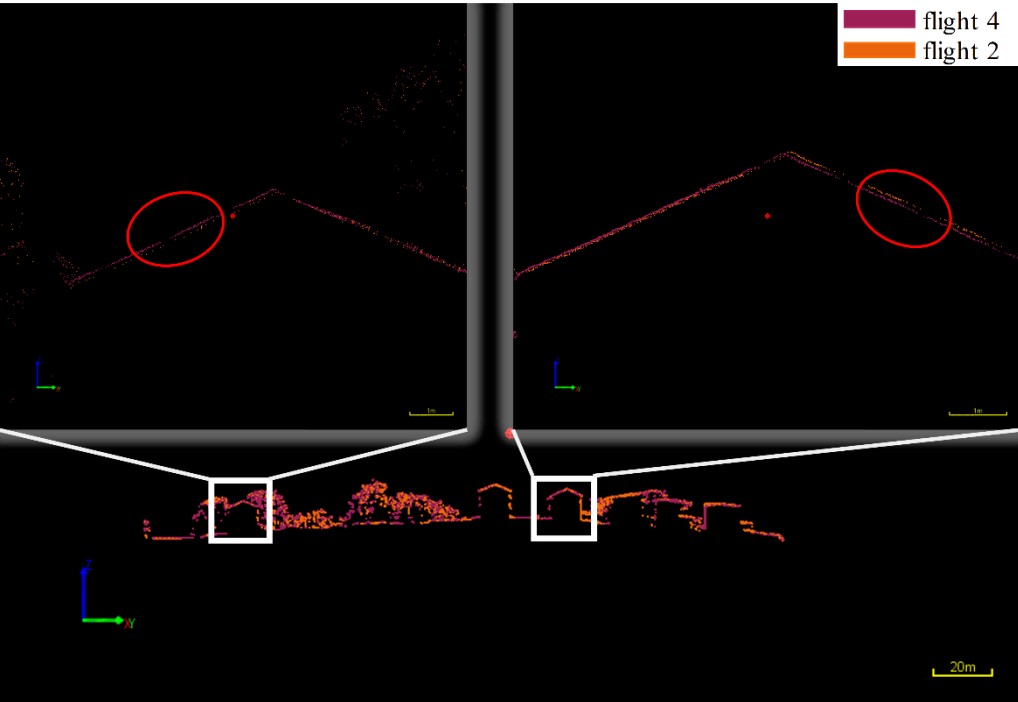

**Figure 21.** Vertical section of point cloud in isotropic flight strips (flight 2 and flight 4) with installation angle compensation.

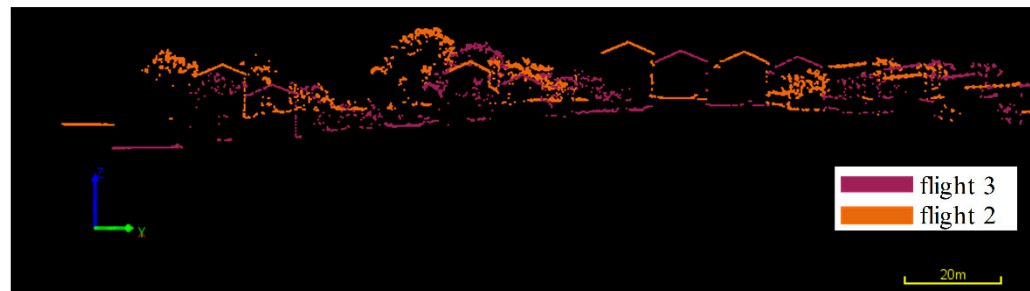

**Figure 22.** Vertical section of point cloud in opposing flight strips (flight 2 and flight 3) without installation angle compensation.

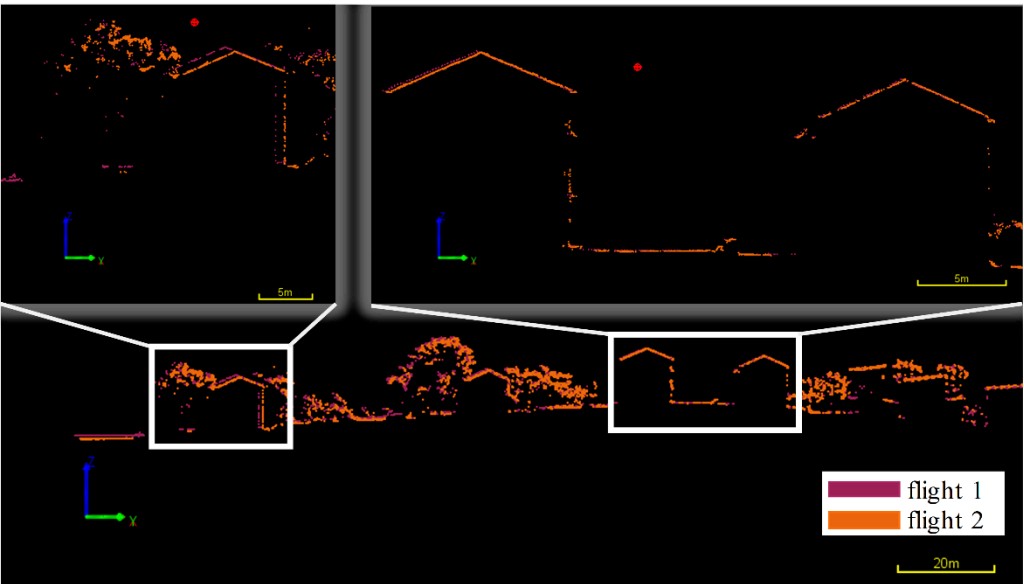

**Figure 23.** Vertical section of point cloud in opposing flight strips (flight 2 and flight 3) with installation angle compensation.

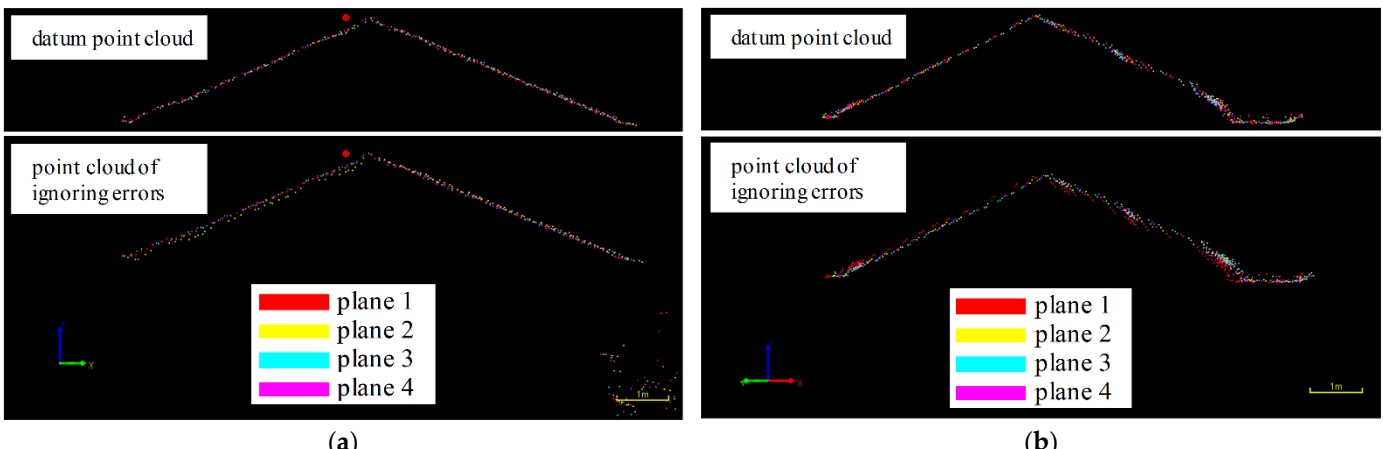

**Figure 24.** Comparison of point clouds with and without surface error compensation. (**a**) The point cloud comparison of the vertical section of the single flight after setting $\Delta\theta_2$, $\Delta\theta_3$, and $\Delta\theta_4$ to 0; (**b**) The point cloud comparison of the parallel section of the single flight after setting $\Delta\varphi_1$, $\Delta\varphi_2$, $\Delta\varphi_3$ and $\Delta\varphi_4$ to 0.

**Table 3.** The elevation difference between the point cloud and RTK control points.

| Number | dz/m | Number | dz/m | Number | dz/m |
| --- | --- | --- | --- | --- | --- |
| pt25 | 0.009 | pt7 | −0.010 | pt10 | −0.027 |
| pt1 | 0.008 | pt15 | −0.012 | pt20 | −0.027 |
| pt3 | 0.008 | pt8 | −0.013 | pt24 | −0.034 |
| pt22 | 0.007 | pt27 | −0.014 | pt12 | −0.036 |
| pt4 | 0.003 | pt19 | −0.015 | pt16 | −0.037 |
| pt6 | 0.001 | pt9 | −0.015 | pt18 | −0.040 |
| pt21 | −0.002 | pt2 | −0.021 | pt11 | −0.065 |
| pt14 | −0.004 | pt26 | −0.025 | pt17 | −0.093 |
| pt23 | −0.006 | pt5 | −0.026 | pt13 | \ |

## 4. Discussion

From the simulation results of the model and the actual flight test results, both the laser and rotation axis parallelism error angles $\omega_z$ and $\omega_y$, and the surface angle deviations $\Delta\theta$ and $\Delta\varphi$ affecting the point cloud cannot be compensated by the installation angle alone.

In the statistics of the angle errors of our LiDARs, the angle errors between the laser and rotation axis ($\omega_z$ and $\omega_y$) are typically $0.13 \pm 0.18°(1\sigma)$ and $0.85° \pm 0.26°$. Systematic errors may be caused by machining and assembly. The surface angle deviations are typically $\pm 0.06°$, which will cause the deviations of about 10 cm/100 m height ($\Delta\theta$) and 20 cm/100 m height ($\Delta\varphi$) in the point cloud when the same target is scanned by different mirrors. The deviations in flight experiment caused by these angle errors are consistent with the simulation results.

As a result, it is necessary to reduce or even eliminate the influence of these angular errors during the production of LiDAR by means of precision installation or calibration. Last but not least, engineers or researchers using LiDAR can also locate the cause of the problem when they encounter poor point cloud alignment due to the angular errors mentioned above.

The following will briefly summarize the effects of the above angular errors:

- The presence of $\omega_z$ will result in a perpendicular flight path, an offset, and an elevation offset. The offsets become larger as the scanning angle increases, and the trend is similar to outward or inward deformation of the point cloud, with respect to the real feature, centered on the flight path.
- The presence of $\omega_y$ will result in three directional offsets between the feature target in the point cloud and its true position. The offset perpendicular to the flight path and elevation and the offset in the elevation direction become larger as the scan angle becomes larger. The trend is similar to that of a point cloud with an overall left or right deformation relative to the real feature centered on the flight path. The offset of the parallel flight path is almost only related to the magnitude of $\omega_y$ but not to the scanning angle.
- The presence of $\Delta\varphi$ will result in a parallel flight path offset between the feature target in the point cloud and its true position, which becomes larger as the scanning angle increases, and the scanning track changes from a straight line to a curve.
- In the presence of $\Delta\theta$ and $\Delta\varphi$, the point clouds of different reflecting surfaces in a single flight strip will be layered.

Furthermore, this method is also applicable to other different scanning systems for improvement as long as it is similar to the scanning model mentioned in this article. For other scanning systems (such as transmission type), the idea presented in this article is also helpful for analysis.

## 5. Conclusions

In this study, we developed the UAV LiDAR, introduced its ranging module and scanning module, and focused on analyzing the mathematical model of its scanning module

and the angular error model. Flight experiments show that the LiDAR operates stably during continuous operation, and the laser transceiver and online waveform processing module can achieve efficient and accurate ranging. In the statistics of the calibrated angle errors of 36 LiDARs of the same type, the angle errors between the laser and rotation axis are typically $0.13 \pm 0.18°$ and $0.85° \pm 0.26°$, and the surface angle deviations are typically $\pm 0.06°$, both of which result in decimeter to meter deviations. After calibration for angular error, the LiDAR can achieve accurate angle measurement. Eventually, the standard deviation of the elevation deviation between the 300 m flight height point cloud and the RTK control point is 0.024 m. The analysis of the experimental results shows that the system has the following technical characteristics:

12    Compact and efficient optomechanical structure

We adopt coaxial design to ensure that laser emission and echo reception share the same optical path. Adopting the quadrangular tower mirror scanning method, the motor obtains four straight uniform scanning lines for every one revolution, which significantly improves the scanning efficiency of the system and reduces the complexity of the overall system structure.

13    Efficient and precise distance measurement

We use 1550 nm laser and narrow band filter to improve the signal-to-noise ratio of the echo signal, nanosecond pulse width laser, low-noise amplifier circuit and ADC with upper GHz sampling frequency to ensure the quality of the echo signal, and online waveform processing to ensure the efficiency and precision of the digital echo signal conversion to range values.

14    Angle errors calibration and compensation

We derived the mathematical model of single-sided mirror reflection and the angular error model and simulated and analyzed the form and law of the influence of the angular error, which was verified in the point cloud of the actual flight experiment. The difference of the point cloud before and after the angular error compensation can be clearly observed.

In summary, the self-developed UAV LiDAR has the characteristics of being long range and having high accuracy, fast measurement frequency, many echoes, fast scan line count, and small volume and weight, which is very suitable for UAV platforms. In this research, we verify the practicality of the self-developed UAV LiDAR and analyze the importance of the mathematical model of the scanning module and the angular error model for accurate measurements. Future work will continue to explore different scanning methods and applications and investigate more practical hardware implementations of range processing algorithms, such as online solving of range ambiguity, which are of great significance for the continuous improvement of UAV LiDAR technology.

**Author Contributions:** Conceptualization, H.Z. and Q.M.; Data curation, A.W.; Formal analysis, H.Z.; Funding acquisition, Q.M.; Investigation, H.Z.; Methodology, H.Z., Y.S. and X.H.; Project administration, H.Z. and A.W.; Resources, Q.M. and A.W.; Software, A.W.; Supervision, H.Z.; Validation, H.Z.; Visualization, H.Z. and X.H.; Writing—Original draft, H.Z.; Writing—Review and editing, H.Z. and Q.M. All authors have read and agreed to the published version of the manuscript.

**Funding:** This research was funded by the National Natural Science Foundation of China (NSFC) under the Grants No 41971413 and U1934215 and was also funded by the research project of China Academy of Railway Sciences Corporation Limited, and the project number is 2021YJ047.

**Data Availability Statement:** Not applicable.

**Conflicts of Interest:** The authors declare no conflict of interest.

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
