# Peer review of "Analysis of Internal Angle Error of UAV LiDAR Based on Rotating Mirror Scanning"

_remotesensing, doi:10.3390/rs14205260_

Round 1

Reviewer 1 Report

Very interesting paper with a lot of technical information about LiDAR internals, which is not usual to find in the literature. The paper provides very interesting conclusions, which can be even used as LiDAR selection criteria or LiDAR device comparision.

I do not understand why flight test are done so high: 300 m, because with UAVs usually flights are under 120 m, and I haven found any explanation related with flights at different heights. Simulation considers 100 m, and higher altitude seems to be a bigger problem.

This paper presents an study on LiDAR data acquisition process and error sources due to the scanning method used in some LiDARs (mirror rotating). Authors compare the characteristics of 45° single-sided mirror scanning, polygonal prism scanning, polygonal tower mirror scanning, and wedge mirror scanning are compared from the single-sided mirror rotation scanning model. An error model is mathematically computed considering the angle deviation between the laser and the rotation axis, the angle deviation between the reflection surface and the rotation axis, and the surface angle deviation between multiple surfaces. Authors base their study on an own produced LiDAR device. The paper present clearly the arquitecture of the system and the sources of error due to the mechanical parts of the LiDAR device.

The paper presents an original simulation model of the error sources due to the construction of LiDAR devices. Such error are relevant to improve calibration process, and to identify which are the best position to deploy the LiDAR device when you flight it within an UAV. This paper clarify for UAV operators how relevant is LiDAR positioning, calibration, and identify source of errors to interpret correctly the point cloud reconstruction results.

The paper presents clearly the arquitecture of LiDAR devices and illustrates very clearly the source of errors. The simulation model is very useful to compare different mirror rotated scanning methods. Additionally, I consider it is a very practical paper because it provides an useful comparison for UAV operators when they have to evaluate LiDAR devices, and could be used as a guide to analyze LiDAR performance during a flight.

It could be interesting to evaluate the model for different flight altitude, at least in simulations.

Conclusions are consistent and address the main question. Conclusions are supported by simulation results and experimental flight tests.

References consider relevant and updated papers related to the topic addressed in the paper.

Figures represent clearly the behavior of mirror rotated scanning methods, and the errors derived due this mode of operation.

Reviewer 2 Report

This is a new and novel paper outlining experimental research and analysis of angular errors in a UAV LiDAR system   The overall analysis, mathematical details, and experimental data are sound and will be used by the Lidar community.   My main suggestions for some improvements are related to better explaining the numerical values of the errors and the effect of optimizing the results.   In the abstract try to indicate the error values normally found in a typical lidar alignment system, the results after optimizing the UAV and lidar alignment, and the improvement for the different scanning systems instead of just stating the error in m at a range of 300m.   This can be also improved in the Discussion section and Conclusions (give typical values for the parameters).
